# XAI-Based Detection of Adversarial Attacks on Deepfake Detectors

**Ben Pinhasov**[*]                                                              *ben.pinhasov@s.afeka.ac.il*
*Afeka, The Academic College of Engineering in Tel Aviv*

**Raz Lapid**[*]                                                                  *razla@post.bgu.ac.il*
*Ben-Gurion University of the Negev, Beer-Sheva, 8410501, Israel & DeepKeep, Tel-Aviv, Israel*

**Rony Ohayon**                                                                   *rony@deepkeep.ai*
*DeepKeep, Tel-Aviv, Israel*

**Moshe Sipper**                                                                  *sipper@bgu.ac.il*
*Ben-Gurion University of the Negev, Beer-Sheva, 8410501, Israel*

**Yehudit Aperstein**                                                             *apersteiny@afeka.ac.il*
*Afeka, The Academic College of Engineering in Tel Aviv*

**Reviewed on OpenReview:** *https://openreview.net/forum?id=7pBKrcn199*

## Abstract

We introduce a novel methodology for identifying adversarial attacks on deepfake detectors using eXplainable Artificial Intelligence (XAI). In an era characterized by digital advancement, deepfakes have emerged as a potent tool, creating a demand for efficient detection systems. However, these systems are frequently targeted by adversarial attacks that inhibit their performance. We address this gap, developing a defensible deepfake detector by leveraging the power of XAI. The proposed methodology uses XAI to generate interpretability maps for a given method, providing explicit visualizations of decision-making factors within the AI models. We subsequently employ a pretrained feature extractor that processes both the input image and its corresponding XAI image. The feature embeddings extracted from this process are then used for training a simple yet effective classifier. Our approach contributes not only to the detection of deepfakes but also enhances the understanding of possible adversarial attacks, pinpointing potential vulnerabilities. Furthermore, this approach does not change the performance of the deepfake detector. The paper demonstrates promising results suggesting a potential pathway for future deepfake detection mechanisms. We believe this study will serve as a valuable contribution to the community, sparking much-needed discourse on safeguarding deepfake detectors.

## 1 Introduction

Deepfake technology, which involves generating realistic media content, has advanced significantly in recent years, leading to the creation of increasingly sophisticated and convincing fake videos, images, and audio recordings (Westerlund, 2019; Khanjani et al., 2021; Guarnera et al., 2020b). In response, there has been a growing interest in developing deepfake detectors capable of identifying and flagging these manipulated media (Le & Woo, 2023; Hou et al., 2023; Lapid et al., 2024a; Rana et al., 2022; Lyu, 2020; Dolhansky et al., 2020; Guarnera et al., 2020a). However, as with any security system, these detectors are vulnerable to

---

[*]Equal contribution
[†]Code available at https://github.com/razla/XAI-Based-Detection-of-Adversarial-Attacks-on-Deepfake-Detectors
[‡]This research was supported by the Israeli Innovation Authority through the Trust.AI consortium

adversarial attacks, which aim to deceive or manipulate detector outputs by making subtle changes to the input, highlighting the need for robust detection methods.

Adversarial examples can significantly compromise the reliability of deepfake detection systems and have the potential to cause harm, especially given the difficulty of detecting deepfakes in general. Therefore, there is a critical need to address the vulnerability of deepfake detectors to adversarial inputs and incorporate defenses against such attacks in the training of detection systems (Hussain et al., 2022).

Concomitantly, the field of *eXplainable Artificial Intelligence (XAI)* has gained momentum, its aim being to render machine learning (ML) and deep learning (DL) techniques more transparent, providing clear interpretations of model decisions to humans.

Previous research has focused on the development of deepfake detection models, with some studies incorporating XAI tools to evaluate the vulnerability of deepfake detectors (Gowrisankar & Thing, 2024). Additionally, there have been efforts to detect deepfake audio using XAI models such as LIME, SHAP, and GradCAM (Govindu et al., 2023).

Despite these advancements, there remains a gap in the research concerning the detection of adversarial attacks on deepfake detectors using XAI-based approaches. The problem area of interest for this paper is the potential vulnerability of deepfake detectors to adversarial attacks and the need for effective adversarial detection mechanisms.

In this paper, we address the aforementioned gap by proposing an XAI-based adversarial detector (Baniecki & Biecek, 2024) for adversarial attacks on deepfake detectors. We conduct experiments to test the effectiveness of the proposed detector in identifying and mitigating adversarial attacks on existing deepfake detection models. By leveraging the insights provided by XAI maps, we aim to improve the ability of deepfake detector systems to accurately distinguish between real and fake content, even if they were adversarially attacked.

Our paper's premise is that deepfake detectors are vulnerable to adversarial attacks and require an additional layer of protection—which we obtain through XAI maps—to enhance their reliability. We seek to examine the following research question:

> **(Q)** How can XAI be used to detect adversarial attacks on deepfake detectors?

To answer **(Q)** we will explore the various XAI methods for detecting such attacks on deepfake detectors.

Our contributions are as follows:

- Introduction of an innovative approach to identifying adversarial attacks on deepfake detection systems. The proposed method offers significant potential for bolstering the security of deepfake detectors and other machine-learning frameworks.

- Integration of XAI techniques to enhance transparency and interpretability in detecting adversarial attacks on deepfake detectors. This addition not only detects attacks but also might provide insights into the decision-making process, fostering trust in the detection outcomes, crucial for real-world applications.

- Empirical evidence demonstrating our method's robustness, through its successful defense against both familiar and previously unseen adversarial attacks, highlighting its resilience and versatility in a variety of adversarial contexts.

The next section describes the previous work done in this area. Section 3 describes the methodology used in the experiments. Section 4 describes our experimental setup and Section 5 presents the results. Our findings are discussed in Section 6, followed by conclusions in Section 7.

## 2 Previous Work

In the domain of deepfake detection, the burgeoning threat of AI-generated manipulations has spurred the development of various detection methodologies. These methodologies broadly fall into two categories: conventional (Le et al., 2022) and end-to-end approaches. Conventional methods, as demonstrated in Figure 1, primarily focus on classifying cropped face images as real or fake. However, their efficacy is contingent upon accurate face detection, leading to vulnerabilities when face localization is imprecise. Notably, recent studies have highlighted the susceptibility of these methods to adversarial manipulations, showcasing the need for enhanced robustness (Hussain et al., 2020).

Figure 1: Conventional approach to deepfake detection: External face detection detects the face crops and each crop passes through the deepfake detector. Each row represents a different adversarial attack.

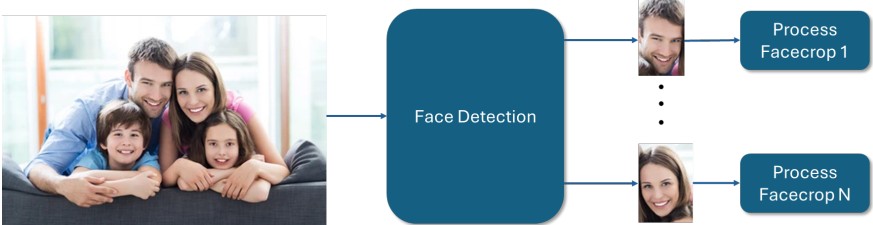

Conversely, end-to-end approaches offer a more comprehensive analysis by not only discerning the authenticity of faces but also localizing manipulated regions at varying levels of granularity, namely detection and segmentation (Le et al., 2022). Despite their advancements, these techniques are not immune to adversarial subversions, underscoring the criticality of bolstering their resilience.

Transitioning to adversarial attacks, the field of machine learning security confronts the persistent challenge posed by adversarial examples. These intentionally crafted inputs exploit vulnerabilities in deep neural network models across diverse domains, compelling erroneous predictions (Szegedy et al., 2014). Noteworthy are the gradient-based attacks, extensively explored in the literature, which exploit gradients to generate adversarial perturbations (Papernot et al., 2015; Lapid & Sipper, 2023a; Papernot et al., 2017; Li & Lyu, 2019; Moosavi-Dezfooli et al., 2017; Tamam et al., 2023; Carlini & Wagner, 2017a; Lapid & Sipper, 2023b; Eykholt et al., 2018; Shi et al., 2019; Lapid et al., 2022; Carlini & Wagner, 2018; Qin et al., 2019; Neekhara et al., 2019; Ebrahimi et al., 2018; Lapid et al., 2024b; Belinkov & Bisk, 2018; Hagiwara et al., 2019).

Recently, Gowrisankar & Thing (2024) presented a novel attack on deepfake detection systems, involving the addition of noise to manipulated images in visual concepts that significantly influence the classification of authentic images, as indicated by a XAI map of the authentic image. In our study we investigated attacks that perturb *all* pixels in the image, rather than confining the perturbation to a limited region.

In practical terms, adversarial attacks on deepfake detectors can lead to dire ramifications, such as the proliferation of misinformation, erosion of trust in media sources, and potentially catastrophic social and political consequences. The surreptitious nature of deepfakes, compounded by their ability to evade detection through adversarial attacks, underscores the urgent need for robust defense mechanisms.

To mitigate such threats, the integration of eXplainable Artificial Intelligence (XAI) emerges as a promising avenue (Angelov et al., 2021; Confalonieri et al., 2021; Došilović et al., 2018; Samek & Müller, 2019). As complex ML models increasingly obfuscate their decision-making processes, XAI endeavors to demystify these black boxes, furnishing interpretable explanations for model predictions (Adadi & Berrada, 2018). By bridging the chasm between AI performance and human comprehension, XAI not only engenders user trust but also enhances accountability and decision-making.

In the domain of deepfake detector defense, existing strategies primarily revolve around fortifying model training to bolster adversarial resilience. Techniques such as adversarial training entail augmenting the training process with adversarial examples to enhance model robustness (Madry et al., 2019). While this approach shows promise, it can inadvertently alter the model's decision boundaries, potentially leading to overfitting or decreased performance on clean data.

Similarly, methodologies like smart watermarking introduce additional complexity to the training pipeline, which may hinder model generalization and scalability (Lv, 2021). Randomized smoothing (Cohen et al., 2019), another prevalent defense mechanism, seeks to mitigate adversarial vulnerabilities by adding random noise to the dataset during training. While effective in certain scenarios, this technique can obscure genuine features in the data, impairing the model's discriminative capabilities. Moreover, the increased computational overhead associated with randomized smoothing may impede real-time deployment, limiting its practical utility.

Chen et al. (2021) proposed a system designed to combat deepfakes, using a two-step process. First, it identified manipulated areas of a video by analyzing inconsistencies and distortions. Then, it employed this information to reconstruct the original, authentic content. This approach effectively defends against deepfakes, even when the attacker's methods are unknown. This deepfake defense might struggle with complex reconstructions and may be computationally expensive.

Despite these drawbacks, these defense mechanisms represent significant strides in fortifying deepfake detectors against adversarial subversion. However, careful consideration of their trade-offs is imperative to ensure a judicious balance between robustness and performance. This paper aims to navigate these nuances by leveraging XAI methodologies to optimize detector performance, particularly in identifying and thwarting adversarial incursions, while mitigating the adverse effects of conventional defense strategies. Importantly, our approach does not alter the underlying model architecture; instead, it involves training a new adversarial detector on top of the existing framework, ensuring compatibility and scalability across different deepfake detection systems.

## 3 Methodology

Our methodology involves a structured approach to improve adversarial-attack detection on deepfake detectors, integrating benchmark dataset, established detection models, creation of an attacked dataset, incorporation of XAI techniques, and our novel detection model. Figure 3 depicts the flow for classifying a face crop as real or attacked.

We start by outlining the dataset preparation used for deepfake detection evaluation (Section 3.1), followed by a review in Section 3.2 of the deepfake detection models applied in our study. We then describe in Section 3.3 the creation of an attacked dataset using four established attack methods, enhancing the testing scope for our approaches. The use of XAI techniques is detailed in Section 3.4, aimed at increasing the interpretability of the adversarial detection process. Lastly, in Section 3.5 we introduce our adversarial-attack detection model, emphasizing its unique contribution to detecting and mitigating adversarial threats, thereby advancing the capability of deepfake detection systems.

### 3.1 Dataset Preparation

To evaluate the robustness of our methodology, we attack the FF+ dataset (see Section 4) using 4 different attacks: Projected Gradient Descent (PGD) (Madry et al., 2017), Fast Gradient Sign Method (Goodfellow et al., 2014), Auto Projected Gradient Descent (APGD) (Croce & Hein, 2020), Natural Evolution Strategies (NES) (Qiu et al., 2021), and Square Attack (Andriushchenko et al., 2020). Herein we focus on the $\| \cdot \|_\infty$ norm constraint. The adversarial detector was trained using PGD only; the other attacks were used for testing only. For training our model the attack process iteratively applied PGD with a fixed maximum perturbation $\epsilon = 16/255$ to generate manipulated instances, resulting in a dataset twice the original size. Subsequently, interpretability maps were generated across this augmented dataset using XAI techniques outlined in Section 3.4, facilitating the assessment of our approach's sensitivity to manipulated features for discriminating between authentic and manipulated videos.

### 3.2 Deepfake Detection Models

**The XceptionNet** architecture (Chollet, 2017) replaces Inception modules (Szegedy et al., 2017) in convolutional neural networks with depth-wise separable convolutions. Depth-wise separable convolutions first

perform a spatial convolution independently over each channel of the input, then perform a 1x1 convolution to project the output channels onto a new channel space.

Xception assumes that mapping cross-channel correlations and spatial correlations in convolutional neural networks can be entirely decoupled. Chollet (2017) showed that there is a spectrum between regular convolutions and depth-wise separable convolutions, with Inception modules being an intermediate point. They demonstrated that Xception, built entirely from depth-wise separable convolutions, achieves slightly better classification performance over the ImageNet dataset as compared to Inception V3 with a similar number of parameters. Chollet (2017) argued that depth-wise separable convolutions offer similar representational power as Inception modules while being easier to use, like regular convolution layers.

In our research we use a pretrained model, as presented by Hussain et al. (2020).

**EfficientNetB4ST** is one model of an ensemble of several models for detecting facial manipulation in videos, presented by Bonettini et al. (2021). EfficientNetB4ST is trained using a Siamese strategy with a triplet margin loss function. This extracts deep features that achieve good separation between real and fake faces in the encoding space. Siamese training produces a feature descriptor that favors similarity between samples of the same class. Bonettini et al. (2021) showed that EfficientNetB4ST complements the other models in the ensemble, demonstrating improved detection accuracy and quality over individual models on the FF++ and DFDC datasets. The fusion of EfficientNetB4ST with other diverse models helps the overall ensemble system outperform the baseline for facial manipulation detection.

In our research, we use a single pre-trained model from the ensemble, which exhibited good performance.

### 3.3 Adversarial Attacks

In this section we describe 5 attack methods that we use for creating the attacked-videos dataset. Three are white-box attacks: PGD (Madry et al., 2017), FGSM (Goodfellow et al., 2014), and Auto Projected Gradient Descent (APGD) (Croce & Hein, 2020). Two are black-box attacks: Natural Evolution Strategies (NES) (Qiu et al., 2021) and Square Attack (Andriushchenko et al., 2020).

White-box attacks involve full access to the target model's architecture and parameters, enabling adversaries to craft adversarial examples more efficiently. In contrast, a black-box attack means an adversary has limited access to the target model and its parameters, making it challenging to craft adversarial examples directly. In the context of image classifiers, an adversarial attack aims to generate perturbed inputs that can mislead the classifier, without explicit knowledge of the model's internal parameters. Black-box attacks have received increased attention in recent years (Andriushchenko et al., 2020; Qiu et al., 2021; Lapid et al., 2022; Chen et al., 2020; Tamam et al., 2023; Lapid & Sipper, 2023a).

We tested our algorithm on various attack algorithms, to assess the robustness and generalization of our method. The attack algorithm PGD was used for creating a training dataset and a test dataset. The attack algorithms APGD, NES, and Square were used for creating a test dataset to assess the robustness of our method.

**PGD**. We assume that the attacker has complete access to the deepfake detector, including the face-extraction pipeline and the architecture and parameters of the classification model. To construct adversarial examples we used PGD (Madry et al., 2017) to optimize the following loss function:

$$\mathcal{L} = \max(Z(x)_{\texttt{real}} - Z(x)_{\texttt{fake}}, 0). \tag{1}$$

Here, $Z(x)_y$ is the logit of label $y$, where $y \in \{\texttt{real}, \texttt{fake}\}$. Minimizing the above loss function maximizes the score for our target label Real. The loss function adopted in this study, as advocated by Carlini & Wagner (2017b), was specifically chosen for its empirical effectiveness in generating adversarial samples with reduced distortion, and its demonstrated resilience to defensive distillation strategies. We used PGD to optimize the above objective while constraining the magnitude of the perturbation as follows:

$$x_{i+1} = x_i - \Pi_\epsilon(\alpha \cdot sign(\nabla_{x_i}\mathcal{L}(\theta, x_i, y))), \tag{2}$$

where $\Pi_\epsilon$ is a projection of the step to $\epsilon$ such that $\|x_i - x\|_p \leq \epsilon, \forall i$, $\epsilon$ is the allowed perturbation, $\alpha$ is the step size, $\mathcal{L}$ is the loss function, $\theta$ is the model's weights, and $y$ is the target prediction.

**FGSM**. The FGSM method, pioneered by Goodfellow et al. (2014), operates by perturbing input data based on the sign of the gradient of the loss function with respect to the input. This perturbation is scaled by a small constant, $\epsilon$, to ensure that changes are imperceptible but effective in fooling the deepfake detector. FGSM is simple and efficient, making it a popular choice for crafting adversarial examples. Despite its simplicity, FGSM can achieve significant perturbations, underscoring its effectiveness in bypassing deepfake detection systems.

**APGD**. The APGD approach, described by Croce & Hein (2020), addresses problematic issues by partitioning N iterations into an initial exploration and an exploitation phase. The transition between these phases involves a gradual step-size reduction. A larger step size allows swift exploration of the parameter space (S), while a smaller one optimizes the objective function locally. Step-size reduction depends on the optimization trend, ensuring it aligns with objective-function growth. Unlike traditional PGD, the APGD algorithm adapts step sizes based on the budget and optimization progress. After step-size reduction, optimization restarts from the best-known point.

**Natural Evolution Strategies (NES)** (Qiu et al., 2021) is an optimization algorithm inspired by the process of natural selection. It operates by iteratively perturbing candidate solutions and selecting those that result in improved fitness. NES does not require gradient information from the black-box model.

**Square Attack** (Andriushchenko et al., 2020) is a black-box attack method specifically designed for image classifiers. It formulates the adversarial example generation as an optimization problem. Square Attack uses random square perturbations because they were proved to successfully fool convolution-based image classifiers, with a smart initialization.

### 3.4 XAI Techniques

**Integrated Gradients** method (Sundararajan et al., 2017) is an attribution technique that aims to identify the input features that have the most significant impact on a deep network's output. To apply this method, the gradient of the output with respect to the input features is integrated along a straight-line path from a baseline input to the actual input. The result provides an attribution value that reflects the contribution of each input feature to the output.

**Saliency**. Simonyan et al. (2013) introduced the concept of saliency as one of the earliest pixel-attribution techniques. This approach involves computing the gradient of the loss function for a specific class of interest with respect to the input pixels. The resulting map shows the relative importance of input features, with negative and positive values indicating their contribution towards or against the class prediction.

**Input** $\times$ **Gradient** technique (Shrikumar et al., 2016) calculates the contribution of input features to a model's output by computing the gradient of the output with respect to each input feature. Specifically, the absolute value of each gradient is multiplied by the corresponding input value to measure the feature's impact on the output. This approach is based on the intuition that higher absolute gradient values indicate more significant contributions to the model's prediction. Overall, the Input x Gradient method provides a means of attributing model outputs to their underlying input features, aiding in model interpretability and identifying potential sources of bias or error.

**Guided Backpropagation** (Springenberg et al., 2014) is a modified version of the backpropagation algorithm that restricts the flow of gradients to only positive values during the backpropagation process. This is achieved by zeroing out the gradients for all negative values. The underlying concept is that positive gradients correspond to input features that positively contribute to the output, while negative gradients indicate features that negatively affect the output. By limiting the flow back to the input to only positive gradients the method aims to highlight the relevant input features that play a vital role in the model's prediction. Consequently, this approach can aid in the interpretation of a network's decision-making process by providing insight into the features that contribute to its output.

Examples of the above XAI techniques can be seen in Figure 2.

Figure 2: FF+ dataset images. Leftmost image: image in pixel space. Four images to the right are XAI maps produced by: Guided Backpropagation (Guid. Back.), Input × Gradient (Inp. × Grad.), Integrated Gradients (Int. Grad.), and Saliency. Top row shows the XAI methods with no attacks, the other rows show the XAI methods for the various attack algorithms. We can clearly see that the XAIs act differently for each given example.

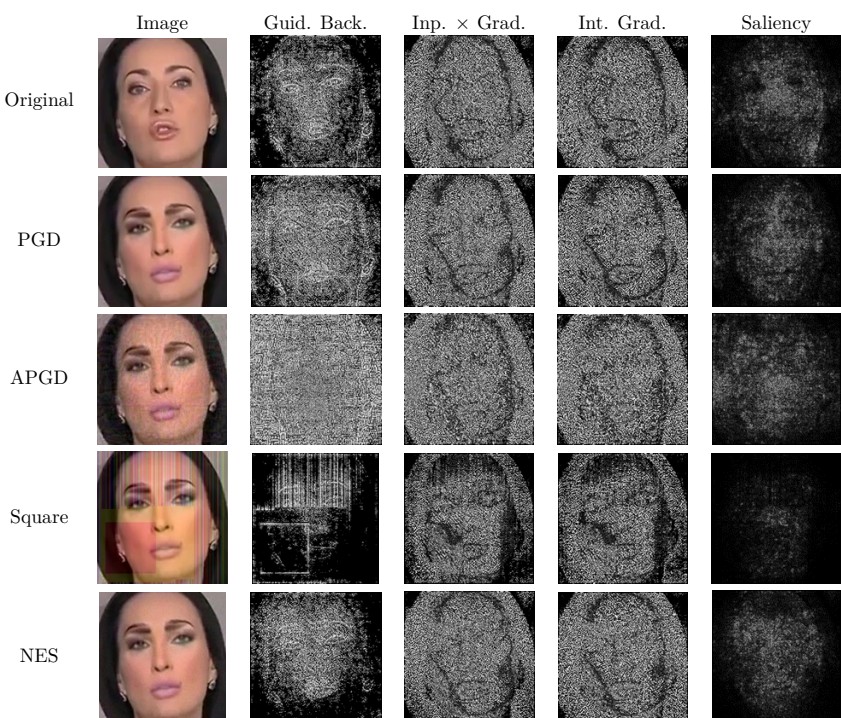

## 3.5 Adversarial Attack Detection

For the creation of our datasets we employed both real and fake videos from the FF+ dataset, creating sets of real or attacked videos and XAI maps. We then formed pairs of unattacked/attacked images and their corresponding XAI maps. Leveraging a pretrained backbone model, e.g., ResNet50 (He et al., 2016), we generated embeddings (feature vectors) for these images. To classify them into unattacked or attacked categories, we applied two layers of linear transformation followed by an activation function. This composite architecture is herein referred to as the Detect-ResNet50 model.

Our detection system operates in the manner illustrated in Figure 3. First, delineate three pivotal components: face detector $f$, deepfake detector $g$, and adversarial detector $d$. Let us define $f : \mathbb{R}^d \to \mathbb{R}^{d'}$ as a face detector extracting the face from a given input image. Subsequently, let $g : \mathbb{R}^{d'} \to \mathbb{R}^2$ denote a binary classifier that classifies an image as deepfake or authentic, based on a face crop. Lastly, $d : \mathbb{R}^{d'} \times \mathbb{R}^{d'} \to \mathbb{R}^2$ represents a binary adversarial detector, evaluating whether a given face crop is under attack, using both the face crop and its corresponding XAI map.

Note that we now have two "axes" of interest, as it were: real vs. fake images, and attacked vs. unattacked images.

Given an image $x \in \mathbb{R}^d$, we begin by passing it through the face detector ($f$), yielding a face crop $x' \in \mathbb{R}^{d'}$. Subsequently, this face crop undergoes assessment by the deepfake detector $g$, producing a classification outcome. In case $g$ classifies $x'$ as authentic, we extract its associated XAI map, $h' \in \mathbb{R}^{d'}$, from $g$. We then submit both $x'$ and $h'$ to the adversarial detector $d$, which in turn determines whether $x'$ has been subjected to an adversarial attack or not.

## 4 Experiments

We conducted experiments to assess the performance of the deepfake detectors based on XceptionNet and EfficientNetB4ST, following the methods described by Chollet (2017) and Bonettini et al. (2021). We used the FaceForensics++ (FF++) dataset (Rossler et al., 2019) to assess the effectiveness of our proposed face-

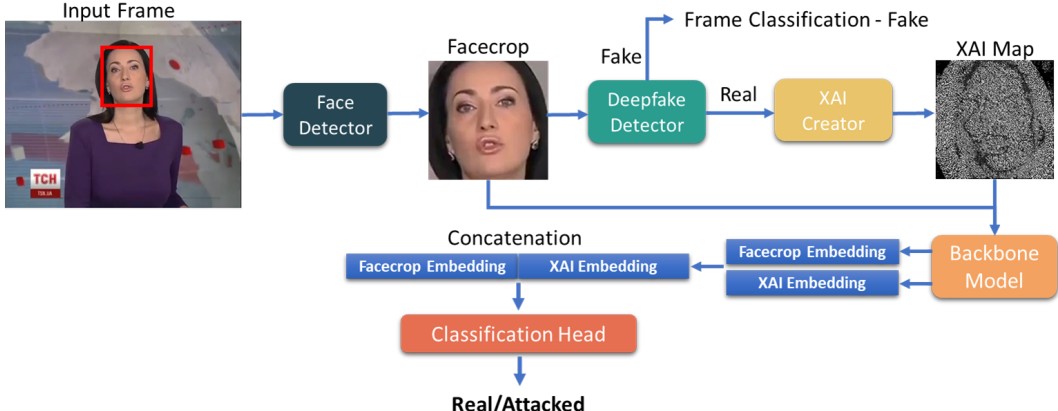

Figure 3: Frame analysis in suspected deepfake videos involves face extraction and classification using deepfake detectors. If classified as fake, the frame is labeled as fake. If classified as real, the face undergoes XAI map creation. The resulting XAI map and face are processed through a backbone model to generate embeddings, which are then input into the classification head to determine 'unattacked' or 'attacked' status.

forgery detection method. This dataset comprises over 5000 videos manipulated using techniques such as DeepFakes, Face2Face, FaceShifter, FaceSwap, and NeuralTextures.

The FF++ dataset consists of 1000 real videos and 1000 fake videos for each manipulation technique. For our experiments, we randomly picked 60 videos from this dataset, with 50 allocated for training (including validation) and 10 for testing, resulting in 20,000 images for training and 5,000 images for testing, per each model and XAI configuration. These videos were subjected to h.254 compression at a compression rate of 23, preparing them for deepfake detection using the models outlined in Section 3.2. The real and fake videos were obtained from the original dataset, while the attacked videos were generated from the fake videos using the techniques presented in Section 3.3.

Herein we focus on the $\|\cdot\|_\infty$ norm constraint. For all attacks conducted we use a perturbation constraint of $\epsilon = 16/255$. In the case of white-box attacks, the hyperparameters used were as follows: for the PGD attack—a maximum of 100 iterations; for FGSM—default configuration (1 iteration); for the APGD attack—a maximum of 100 iterations and 5 restarts. For black-box attacks the hyperparameters were: for the NES attack—a step size of $1/255$, a maximum of 100 iterations, and 5 restarts; for the Square attack—a maximum of 5000 iterations, and the sampling-distribution parameter was set to 0.8.

We evaluated deepfake detection accuracy using two metrics. The *videos* metric, denoted as `Vid`, computes the majority vote of all fake and real frames in each video and assigns a label of fake if more than 50% of the frames were fake. The *frame-by-frame* metric, denoted as `F2F`, measures the average precision of all individual frames.

Table 1 summarizes the pre-attack detection results, i.e., before performing adversarial attacks.

Table 1: Precision of deepfake detectors on FF+ dataset. Precision was calculated by 2 metrics: video and frame-by-frame (see text).

|  | Real | | Fake | |
|---|---|---|---|---|
|  | Vid | F2F | Vid | F2F |
| XceptionNet | 100.00% | 99.36% | 100.00% | 99.82% |
| EfficientNetB4ST | 96.00% | 88.59% | 98.00% | 96.55% |

Table 2 summarizes the post-attack detection results on deepfake images. The latter shows that attacked images cause the deepfake detector to lose its ability to ascertain real from not.

Table 2: Precision of deepfake detectors on the attacked FF+ dataset. Herein, we focus only on attacked deepfake images that are supposed to be classified as deepfake. When attacked, the performant deepfake detectors of Table 1 fail completely. This shows that attacked images cause the deepfake detectors to lose their ability to differentiate real from not.

|  | EfficientNetB4ST | | Xception | |
|---|---|---|---|---|
|  | Vid | F2F | Vid | F2F |
| PGD | 0.00% | 0.00% | 0.00% | 1.00% |
| FGSM | 20.00% | 20.66% | 0.00% | 0.00% |
| APGD | 0.00% | 0.10% | 0.00% | 0.10% |
| Square | 0.00% | 0.20% | 0.00% | 0.10% |
| NES | 0.00% | 5.70% | 0.00% | 1.00% |

The Detect-ResNet50 model underwent training using two distinct configurations. First, the finetuning process involved both the ResNet50 backbone model and the associated classification head. In the second configuration, the backbone model remained in a frozen state, and training was administered only to the classification head. The schematic of the Detect-ResNet50 model is presented in Figure 4.

Figure 4: Detect-ResNet50 architecture. The backbone model contains pretrained ResNet50, and the classification head contains 2 linear layers.

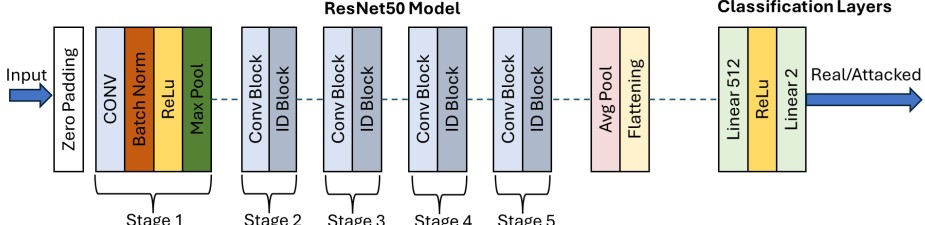

Detect-ResNet50 model was trained with a learning rate of 0.001, a minibatch size of 16, and 100 iterations. We used the cross-entropy loss function, and the Adam optimizer (Kingma & Ba, 2014)—with hyperparameters $\epsilon = 1e - 8$, $\beta_1 = 0.9$, and $\beta_2 = 0.999$.

Furthermore, we conducted a comparison of the dataset's accuracy under two conditions: 1) when using XAI maps, and 2) when these maps are rendered black (resulting in zero tensors), referred to as PGD-B. This latter comparison aims to show the importance of XAI maps on accuracy and to establish a baseline for testing. Specifically, we employed trained models and tested them without the presence of XAIs.

In addition, our study incorporates adaptive attack strategies into the evaluation framework. Importantly, our proposed detection methodology assumes the adversary does not possesses simultaneous access both to deepfake detection and to adversarial detection mechanisms. We believe this assumption accords with real-life scenarios—such as those encountered in social media platforms, where an attacker's actions are limited to uploading a video without knowledge of the underlying modules operating within the application. We advocate for the deployment of our approach in real-world settings, emphasizing its accessibility exclusively through API calls. Herein, we used a simple loss function:

$$\mathcal{L}_{\text{standard-adaptive}} = \mathcal{L}_{\text{BCE}}(f(x + \delta), \texttt{real}) + \mathcal{L}_{\text{BCE}}(g(x + \delta), \texttt{unattacked}), \quad (3)$$

where BCE is the Binary Cross-Entropy Loss, $f$ is the deepfake detector, $\texttt{real}$ is the 'real' class, $g$ is the adversarial detector, and $\texttt{ben}$ is the 'benign' class. Thus the adversary's goal is to minimize $\mathcal{L}_{\text{standard-adaptive}}$.

Moreover, we added another loss function, which tries to minimize the distance of the resultant XAI map:

$$\mathcal{L}_{\text{XAI-adaptive}} = \mathcal{L}_{\text{BCE}}(f(x + \delta), \texttt{real}) + \|(\text{XAI}(x + \delta; f), \text{XAI}(x; f))\|_2, \quad (4)$$

where XAI is a function that takes an input and a model, and outputs a XAI map. Again, the adversary's goal is to minimize this loss function.

Finally, we incorporated generalization assessments into our study, wherein we evaluated the efficacy of an adversarial detection model initially trained on XceptionNet against the alternative architecture of Efficient-NetB4ST, and, conversely, we examined the performance of the model trained on EfficientNetB4ST when confronted with XceptionNet.

## 4.1 Computational Overhead Experiments

To comprehensively assess the computational overhead introduced by integrating XAI techniques, we designed an experiment specifically focusing on measuring the additional computational costs. The experimental setup included a high-performance computing environment with an Intel Xeon Gold 6136 @ 3Ghz CPU, NVIDIA GRID P40-24Q, and 64GB of RAM. The software environment consisted of Windows 10, Python 3.9, Pytorch 2.0.1, and the Captum library for implementing XAI techniques (Kokhlikyan et al., 2019) . We measured the processing time and resource usage (CPU/GPU utilization and memory consumption) of our adversarial detection model, both with and without the integrated XAI methods.

The experiment was structured as follows:

- **Baseline Measurement**: Run the adversarial-detection model without any XAI techniques, recording the processing time and memory consumption.

- **XAI Integration Measurement**: Integrate the selected XAI techniques into the pipeline and repeat the measurements.

- **Comparison**: Compare the baseline and XAI integration measurements to quantify the overhead.

## 5 Results

Table 3 and Table 4 assess the efficacy of our approach against adversarial attacks on deepfake detectors. We focus on two distinct scenarios: one with a finetuned ResNet50 model and the other with a frozen pretrained ResNet50 model, finetuning the classifier head only. We set herein a classification threshold of 0.5; Appendix A presents statistics about different thresholds.

Table 3: Accuracy (Real/Attacked in Figure 3) for various attacks, on the complete test set of a finetuned ResNet50. The asterisk (*) associated with the PGD attack denotes its use in training the model. The best average result reached is shown in boldface. Mean does not include PGD-B experiments.

| Deepfake Detector | Attack | G-Backprop | Inp × Grad | Int-Grad | Saliency |
|---|---|---|---|---|---|
| EfficientNetB4ST | (PGD-B*) | (85.04%) | (84.61%) | (85.00%) | (89.16%) |
| | PGD* | 99.60% | 97.54% | 94.48% | 99.60% |
| | FGSM | 91.90% | 89.93% | 85.62% | 93.11% |
| | APGD | 98.86% | 97.10% | 93.80% | 94.40% |
| | Square | 61.36% | 63.13% | 62.13% | 62.30% |
| | NES | 99.80% | 97.66% | 95.24% | 99.57% |
| **Average** | | **90.30%** | 89.07% | 86.24% | 89.80% |
| XceptionNet | (PGD-B*) | (61.43%) | (81.83%) | (81.81%) | (53.13%) |
| | PGD* | 95.92% | 86.63% | 82.93% | 90.57% |
| | FGSM | 50.88% | 73.67% | 65.22% | 85.73% |
| | APGD | 91.06% | 54.83% | 77.47% | 67.34% |
| | Square | 52.41% | 81.98% | 80.75% | 82.20% |
| | NES | 88.54% | 86.75% | 85.80% | 89.78% |
| **Average** | | 75.76% | 76.77% | 78.43% | **85.12%** |
| **Total Average** | | 83.03% | 82.92% | 82.33% | **87.46%** |

For the finetuned ResNet50 model (Table 3), we observe that the average accuracy across all evaluated attacks ranges from 75.76% to 90.30%. Notably, Saliency exhibits the highest success rate at detecting

Table 4: Accuracy (Real/Attacked in Figure 3) for various attacks, on the complete test set, when finetuning only the classification head of a pretrained ResNet50. The asterisk (*) associated with the PGD attack denotes its use in training the model. The best average result reached is shown in boldface. Mean does not include PGD-B experiments.

| Deepfake Detector | Attack | G-Backprop | Inp × Grad | Int-Grad | Saliency |
|---|---|---|---|---|---|
| EfficientNetB4ST | (PGD-B*) | (69.00%) | (74.65%) | (78.26%) | (78.08%) |
| | PGD* | 93.46% | 92.36% | 91.58% | 98.02% |
| | FGSM | 76.65% | 82.26% | 83.70% | 88.02% |
| | APGD | 96.04% | 95.14% | 92.94% | 98.83% |
| | Square | 53.67% | 61.22% | 60.25% | 65.60% |
| | NES | 93.57% | 91.33% | 88.60% | 96.60% |
| **Average** | | 82.68% | 84.46% | 83.41% | **89.41%** |
| XceptionNet | (PGD-B*) | (50.32%) | (63.13%) | (53.14%) | (65.70%) |
| | PGD* | 90.07% | 69.44% | 70.33% | 72.80% |
| | FGSM | 51.28% | 60.70% | 65.16% | 73.55% |
| | APGD | 93.57% | 73.01% | 69.65% | 82.72% |
| | Square | 46.03% | 57.53% | 53.76% | 59.68% |
| | NES | 81.13% | 76.70% | 76.05% | 82.85% |
| **Average** | | 72.41% | 67.47% | 67.00% | **74.32%** |
| **Total Average** | | 77.54% | 75.96% | 75.21% | **81.85%** |

the attacks, achieving an average accuracy of 87.46%. Conversely, G-Backprop, Inp × Grad and Int-Grad demonstrate relatively lower average accuracies of 83.03%, 82.92% and 82.33%, respectively.

Performance slightly declined when confronted with the Square attack, while using EfficientNetB4ST, with accuracies ranging from 61.36% to 63.13% across all the XAI techniques. This decrease in performance can be attributed to the model's training using PGD only, indicating a potential limitation in generalization to other adversarial attack types. On the other hand, the performance of our approach against Square Attack on XceptionNet is better, reaching a best score of 82.20% with Saliency.

These results underscore the vulnerability of the XAI techniques to various adversarial attacks, with notable variations in attack success rates.

Turning our attention to the second configuration of the ResNet50 model (Table 4), i.e., finetuning the classification head only, we note similar trends in attack effectiveness. Across all evaluated attacks, the average accuracy ranges from 67.00% to 89.41%. Once again, Saliency emerges as the most successful, achieving an average accuracy of 81.85%. This is followed by G-Backprop, with an average accuracy of 77.54%. Conversely, Inp × Grad and Int-Grad exhibit lower average accuracies of 75.96% and 75.21%, respectively.

We note the role of PGD-B in elucidating the significance of XAI in our approach. PGD-B represents a scenario where XAI contributions are nullified, essentially simulating a scenario devoid of interpretability. In this context, PGD-B results underscore the pivotal role of XAI techniques in our methodology. When XAI is absent, the effectiveness of our approach dwindles, (up to ∼ 10% difference in the finetuning experiments and up to ∼ 40% in finetuning the classifier head only), highlighting the dependency on XAI methodologies for robust detection of adversarial attacks on deepfake detectors.

The experimentation underscores the criticality of leveraging XAI not only to enhance model interpretability but also to fortify model resilience against adversarial manipulations, thereby underscoring the synergy between XAI and traditional input features in bolstering deepfake detection mechanisms.

Our findings underscore the balance between finetuning the entire model versus specific layers, adding another consideration. Finetuning the entire model allows for thorough adaptation to the target domain, enhancing task alignment and performance against adversarial attacks. Yet, it requires significant computational resources and may not be viable in resource-limited settings. On the other hand, finetuning only certain layers provides a more practical option, reducing computational burden while enabling some domain adaptation.

However, it restricts model adaptation and may result in lower performance, especially against advanced adversarial attacks. On average, the detection rate on EfficientNetB4ST is better than the detection rate on XceptionNet.

Table 6 and Table 7 show results of the adaptive attacks, which are aware of both the deepfake detector and the adversarial detector, while using Equation 3 and Equation 4 respectively. We emphasize again that we do not think it is a realistic scenario and our detector was not trained against these kinds of attacks. In the first scenario, using Equation 3, the attacks successfully fooled both detectors, with EfficientNetB4ST showsing more resillience than XceptionNet. Using Equation 4, EfficientNetB4ST shows excellent performance, while XceptionNet still fails to resist these attacks.

Table 5: Comparison of Adaptive Attacks

Table 6: Adaptive attacks using Equation 3.

|  | XceptionNet | | EfficientNetB4ST | |
|---|---|---|---|---|
|  | Vid | F2F | Vid | F2F |
| PGD | 0.00% | 0.00% | 0.00% | 0.16% |
| FGSM | 0.00% | 0.49% | 80.00% | 88.50% |

Table 7: Adaptive attacks using Equation 4.

|  | XceptionNet | | EfficientNetB4ST | |
|---|---|---|---|---|
|  | Vid | F2F | Vid | F2F |
| PGD | 0.00% | 0.24% | 100.00% | 98.60% |
| FGSM | 0.00% | 0.25% | 100.00% | 98.30% |

In addition we conducted a generalization study where we evaluated the generalization of our approach: training on one model and evaluating on another. The results are delineated in Table 8, where we see generalization performance ranging from 66.00% to 82.97%. We note that on average training on XceptionNet's attacked images and evaluating on EfficientNetB4ST yields the best results, ranging from 75.05%-82.97%.

Table 8: Transferability results for various attacks. A pretrained adversarial detector trained on a backbone deepfake detector is tested on attacks that were optimized on another backbone detector.

| Source Model → Transferred Model | Attack | G-Backprop | Inp × Grad | Int-Grad | Saliency |
|---|---|---|---|---|---|
| EfficientNetB4ST → XceptionNet | PGD* | 67.33% | 68.03% | 80.50% | 71.68% |
|  | FGSM | 67.38% | 68.02% | 80.67% | 71.67% |
|  | APGD | 70.30% | 63.37% | 82.80% | 85.30% |
|  | Square | 47.10% | 66.56% | 82.93% | 76.45% |
|  | NES | 77.90% | 68.08% | 81.64% | 99.60% |
| **Average** | | 66.00% | 66.81% | **81.71%** | 80.94% |
| XceptionNet → EfficientNetB4ST | PGD* | 95.90% | 86.84% | 83.21% | 90.68% |
|  | FGSM | 50.88% | 73.67% | 65.22% | 85.73% |
|  | APGD | 89.80% | 54.83% | 77.85% | 66.10% |
|  | Square | 52.41% | 81.98% | 80.73% | 82.18% |
|  | NES | 86.35% | 87.25% | 83.32% | 90.16% |
| **Average** | | 75.07% | 76.91% | 78.06% | **82.97%** |
| **Total Average** | | 70.54% | 71.86% | 79.88% | **81.95%** |

Overall, we have assessed the performance of different XAI methods, with varying degrees of success, depending on the attack. Training an adversarial detector on EfficientNetB4ST yields a more-resilient detector than XceptionNet, on average. Our findings emphasize the importance of developing robust defense strategies to safeguard against the proliferation of synthetic manipulations of digital media.

## 5.1 Computational Overhead Results

The results of our computational overhead experiments are presented in Table 9, which summarizes the processing-times metrics for both the baseline model and the pipeline with integrated XAI techniques.

These results indicate that while XAI techniques enhance explainability, they also introduce varying degrees of computational overhead. The Integrated Gradients method, in particular, incurs substantial overhead, making it less practical for real-time applications, compared to other XAI techniques. Conversely, methods like G-Backprop and Input × Gradient offer a more balanced trade-off between computational cost and

Table 9: Computational overhead of our proposed pipeline.

| Model | Average Processing Time (ms) | | | | |
|---|---|---|---|---|---|
| | Baseline | G-Backprop | Inp × Grad | Int-Grad | Saliency |
| XceptionNet | 222.0 | 338.1 (+52%) | 269.4 (+21%) | 3640.3 (+1539%) | 264.6 (+19%) |
| EfficientNet | 247.1 | 361.5 (+46%) | 335.0 (+36%) | 8440.3 (+3316%) | 358.1 (+45%) |

explainability, exhibiting relatively lower processing times while still providing valuable insights into model decisions.

In addition to processing time, we also measured the memory overhead introduced by the XAI techniques. The baseline models have a memory usage of approximately 1.1GB. Integrating the XAI methods increases the memory usage by approximately 1GB, resulting in a total memory consumption of around 2.1GB. This additional memory overhead is primarily due to the storage requirements for the gradients and intermediate computations used by the XAI techniques.

The increased memory footprint is a crucial consideration for deploying these models in real-world settings, especially on resource-constrained devices. While the overhead is significant, it remains manageable within the context of modern computing environments equipped with sufficient memory resources. This analysis underscores the importance of balancing the need for explainability with the available computational resources, particularly in applications where memory constraints are a critical factor.

# 6 Discussion

**Effectiveness of XAI in adversarial detection**. The use of XAI techniques in the context of detecting adversarial attacks on deepfake detectors has demonstrated significant promise. The interpretability provided by XAI methods, such as feature attribution and saliency maps, enhances our understanding of model decision-making processes. In our experiments the integration of XAI facilitated the identification of subtle adversarial manipulations that might have otherwise gone unnoticed, as shown in the black XAI image experiments (Table 3 and Table 4). This highlights the importance of incorporating explainability mechanisms in deepfake detection systems to improve their robustness against adversarial attacks. Additionally, our findings reveal nuanced insights into the effectiveness of XAI techniques in mitigating adversarial attacks. Specifically, we observed that Saliency and G-Backprop outperform other methods in scenarios where the entire network is fine-tuned, while Saliency demonstrates superior performance when only the classification head is fine-tuned. Furthermore, on average, using EfficientNetB4ST's XAIs yield better results than XceptionNet. This distinction underscores the importance of considering the architectural and training constraints when integrating XAI methods for robust deepfake detection.

**Ethical Use of XAI**. While this paper emphasizes the benefits of using XAI to enhance deepfake detectors, it is crucial to consider potential ethical concerns. XAI could potentially be misused by adversaries to reverse-engineer AI systems and exploit vulnerabilities, thereby compromising the very robustness these techniques aim to enhance. To mitigate such risks, it is essential to implement strict security protocols governing the dissemination and application of XAI insights. This includes controlling access to sensitive interpretability data, collaborating with cybersecurity experts, and developing XAI methods that balance transparency with security. Continuous ethical deliberation within the research community is also necessary to ensure responsible AI usage.

**Societal Implications**. Enhanced deepfake detectors have significant societal benefits, such as preventing misinformation and protecting digital content integrity. However, their dual-use nature raises concerns about privacy and surveillance. These technologies could be misused to monitor individuals' digital expressions, infringing on privacy rights. To address these concerns, comprehensive guidelines for the responsible deployment of deepfake detectors must be proposed and adhered to. These guidelines should ensure transparency, legal and ethical consistency, and protection of privacy rights. Multi-stakeholder dialogue involving policy-

makers, technologists, civil society, and the public is essential to define and enforce ethical boundaries for deploying these technologies.

**Vulnerabilities of deepfake detectors**. Despite advances in deepfake detection models, our study reveals inherent vulnerabilities that adversarial actors can exploit. Adversarial attacks, particularly those crafted with the intent to deceive deepfake detectors, pose a formidable challenge. Traditional evasion techniques, such as input perturbations and gradient-based attacks, were successful in deceiving state-of-the-art deepfake detectors. This underscores the need for ongoing research and development in enhancing the robustness of deepfake detection models against adversarial manipulations.

**Importance of explainability for trustworthiness**. Explainability not only contributes to the detection of adversarial attacks but also plays a crucial role in establishing trustworthiness in AI systems. The ability to provide clear and interpretable justifications for model predictions instills confidence in end-users and facilitates a better understanding of potential vulnerabilities. In applications where the consequences of false positives or false negatives can be severe, the transparency afforded by XAI methods becomes indispensable. Thus, XAI-based methods should be further employed and analyzed.

**Generalization and transferability**. Our experiments considered a diverse set of adversarial attacks and deepfake detection models to evaluate the generalization and transferability of adversarial attacks. Our findings suggest that certain adversarial techniques remain effective across different datasets and models, emphasizing the need for standardized evaluation protocols and robust defenses that can withstand a variety of adversarial strategies.

**Limitations**. While our proposed adversarial detector for attacks on deepfake detectors demonstrates promising results, its efficacy may be limited by the diversity of adversarial attacks it has been trained on, and its generalization capability across different models and datasets. Additionally, exploring a wider range of $\epsilon$ values in crafting adversarial examples could provide deeper insights into the robustness of the detector. Moreover, the possibility of adversaries targeting our detector itself and the ethical implications of its deployment underscore the need for ongoing research to address these challenges and ensure its practical applicability and societal impact. However, it is reasonable to assume that the attacker lacks access to the adversarial detector.

**Future directions**. The dynamic landscape of adversarial attacks on deepfake detectors calls for continuous research efforts. Future directions may include the exploration of novel XAI techniques, the development of adversarially robust deepfake detection models, and the investigation of real-time detection strategies. Additionally, interdisciplinary collaborations involving experts in computer vision, machine learning, and ethics can contribute to a holistic approach in addressing the evolving challenges posed by deepfake technology.

## 7 Conclusions

The use of XAI techniques affords significant potential in identifying adversarial attacks on deepfake detectors, with Guided Backpropagation proving notably accurate in this regard. Enhancing model interpretability is crucial in bolstering detection capabilities.

Deepfake detectors are vulnerable to adversarial attacks, facilitated by methods like PGD, APGD, NES, and Square, emphasizing the need for robust defense mechanisms to maintain reliability and efficacy. While fine-tuning the entire adversarial detection model enhances adaptability and resilience, it entails computational overhead, necessitating careful consideration.

XAI maps offer valuable cues for discerning adversarial perturbations, aiding in the differentiation between authentic and manipulated inputs, thereby reinforcing defenses. Despite promising generalization capabilities, diversification in attack modalities during training is essential for improved efficacy and robustness.

Acknowledging inherent constraints—especially concerning attack diversity and evaluation across models—underscores the necessity for continued research to enhance robustness and generalizability. Further exploration into scenarios where attackers have knowledge of detection systems is crucial for developing countermeasures against sophisticated attacks.

In conclusion, the synergy between XAI and robust learning strategies shows promise in safeguarding deep-fake detectors. Continued research is vital to address limitations and broaden the applicability of these methodologies, emphasizing the importance of collaborative efforts in fortifying AI systems' reliability and integrity.

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

## A   Appendix

We present ROC curves for our experiments on detecting adversarial attacks on deepfake detectors. These curves illustrate the performance trade-offs of our models under different attack scenarios.

The ROC curves provide a visual representation of our models' sensitivity to deepfake detection amidst adversarial manipulation. By examining these curves, readers can assess the effectiveness of our proposed techniques in mitigating adversarial attacks.

Tables Table 10 and Table 11 show the results for a fully finetuned and a classification-head only finetuned ResNet50 on EfficientNetB4ST, respectively. Tables Table 12 and Table 13 show the same results while using XceptionNet as the deepfake detector.

Table 10: RoC curves of a fully finetuned ResNet50 on adversarial attacks that were optimized on Efficient-NetB4ST.

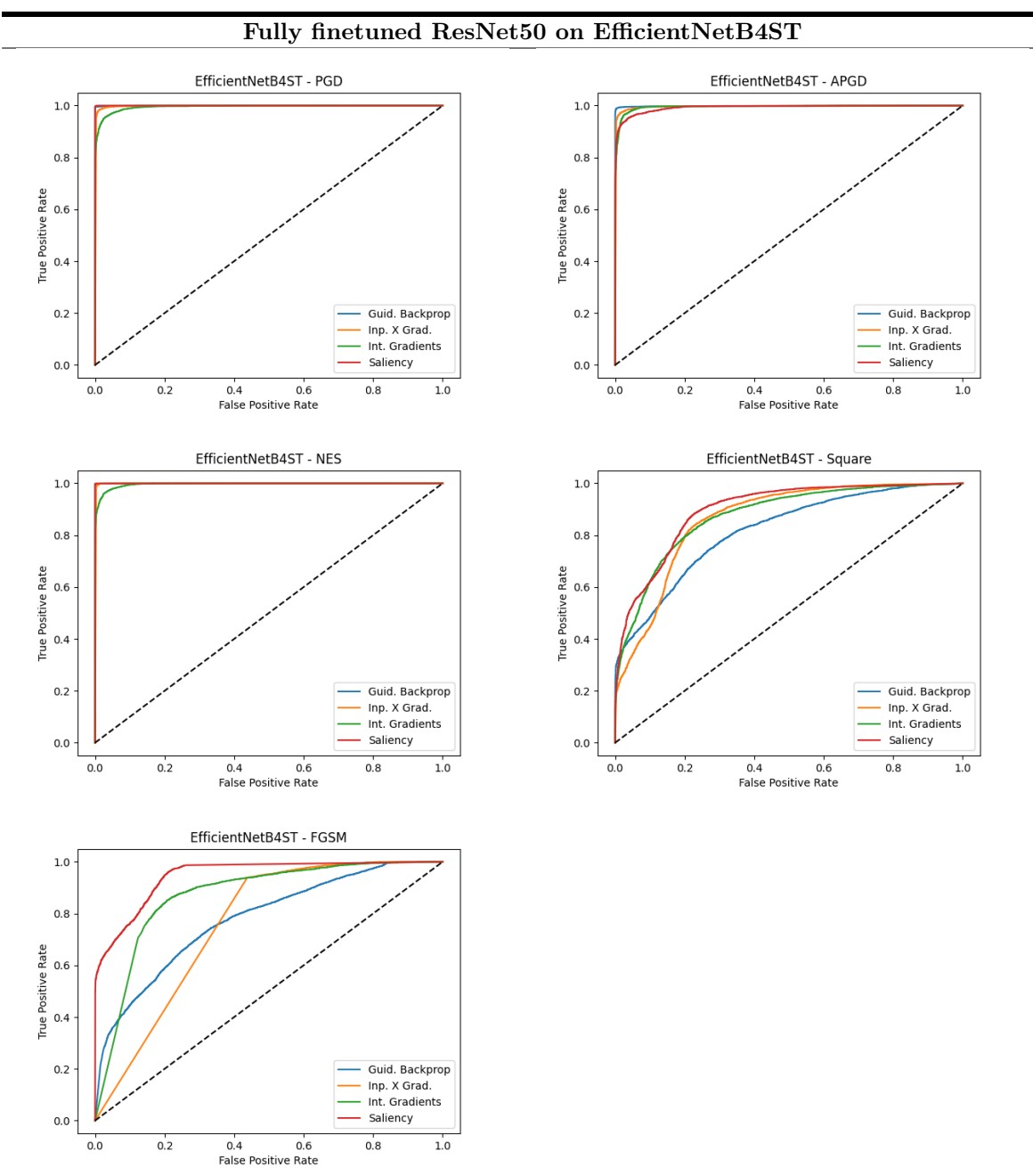

Table 11: RoC curves of a classification head-finetuned ResNet50 on adversarial attacks that were optimized on EfficientNetB4ST.

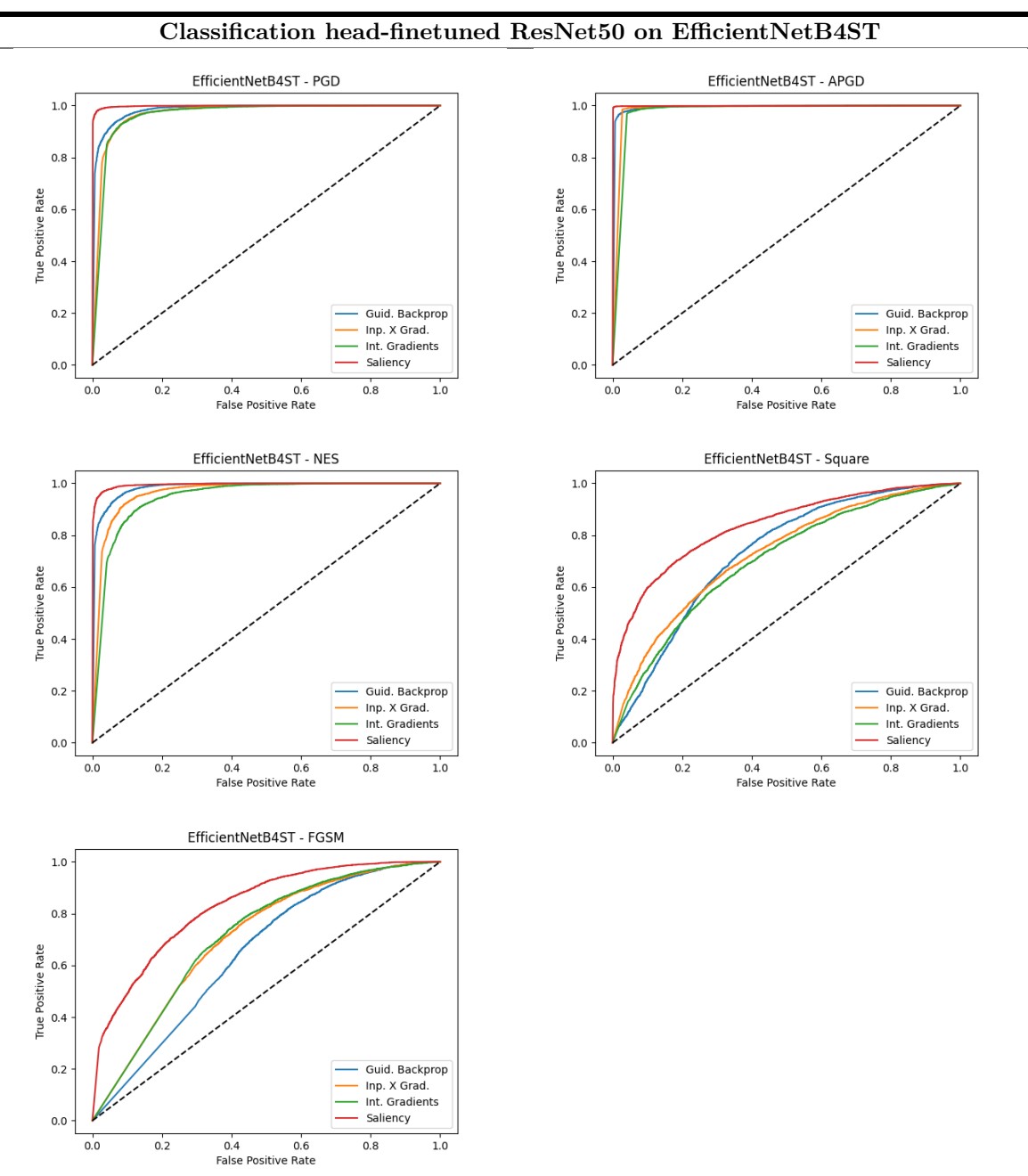

Table 12: RoC curves of a fully finetuned ResNet50 on adversarial attacks that were optimized on Xception-Net.

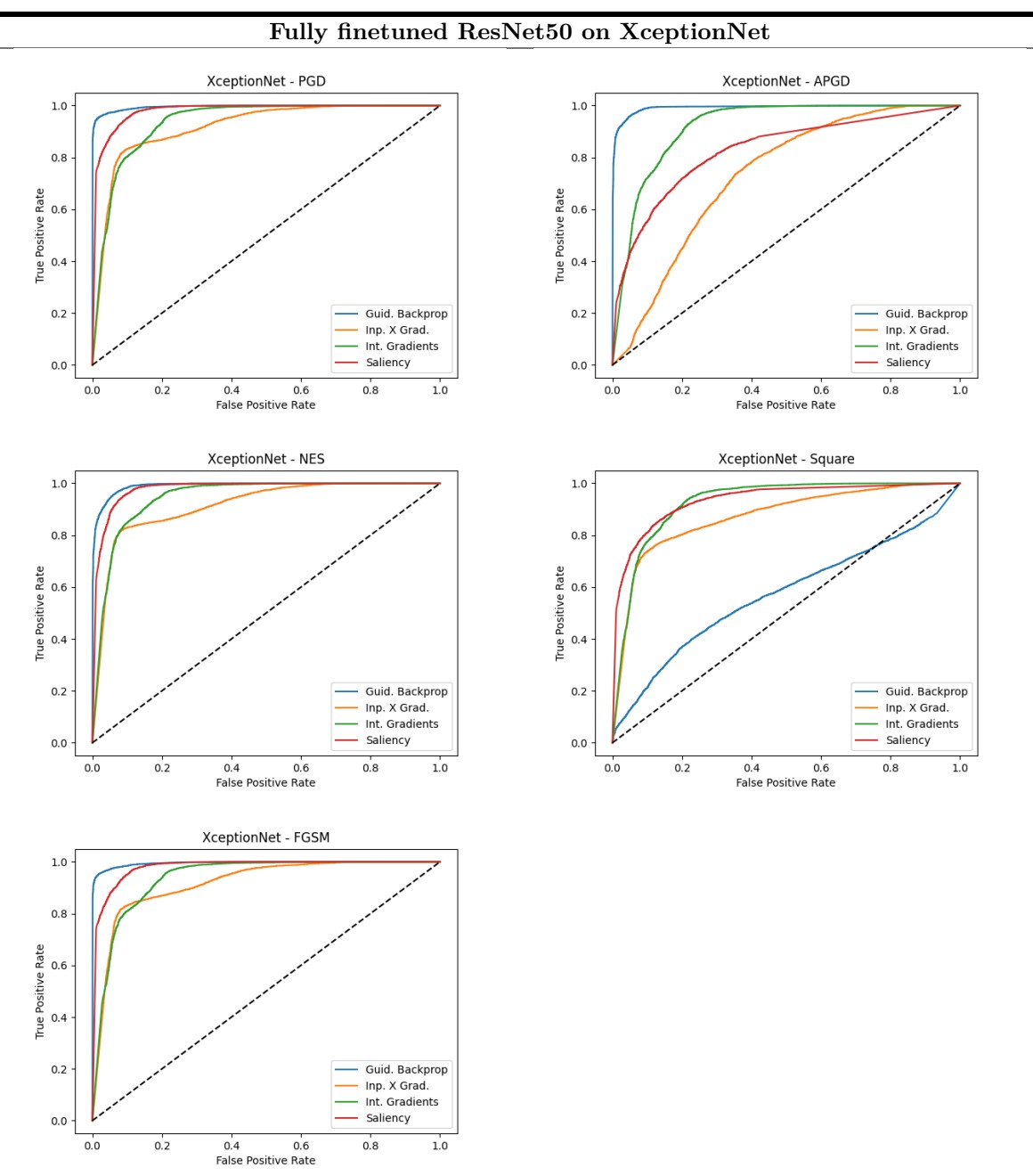

Table 13: RoC curves of a classification head-finetuned ResNet50 on adversarial attacks that were optimized on XceptionNet.

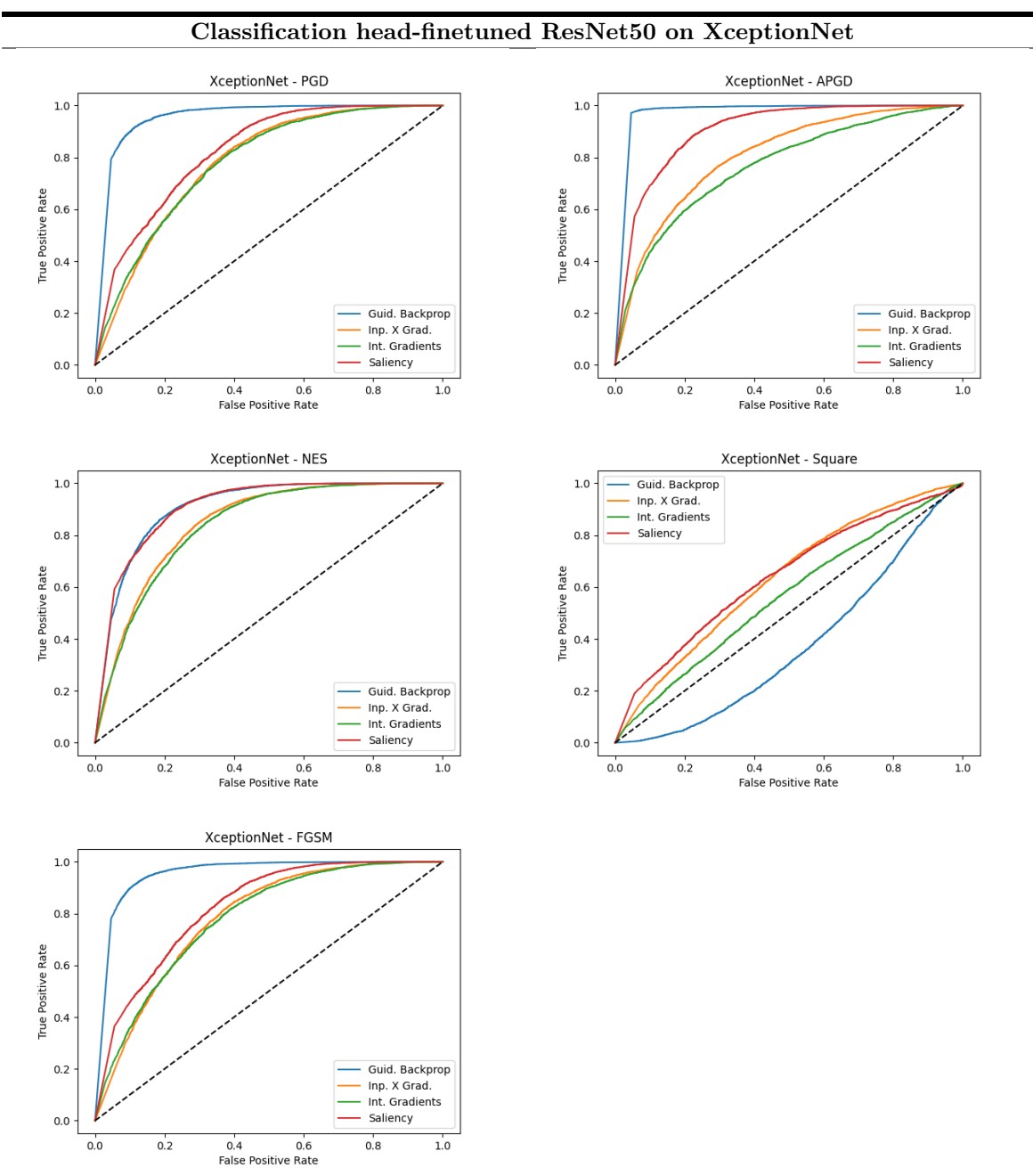

