# OpenReview forum: "XAI-Based Detection of Adversarial Attacks on Deepfake Detectors"
_TMLR — Accepted by TMLR_

### Review · Reviewer_FZic · 2024-03-28

**Summary Of Contributions:**

Detecting deepfake vs. real videos is a challenge to ML. Existing deepfake detectors are prone to adversarial examples, i.e., carefully calibrated deepfake examples can evade the detection. This paper proposes a defense mechanism by introducing XAI features in the detection process. The mechanism performs well on selected attacks as the XAI features before and after the attacks are significantly different.

**Audience:**

Yes

**Claims And Evidence:**

No

**Requested Changes:**

Please list two possible adaptive attacks to the current defense mechanism and report the robust accuracy against these stronger attacks. Thanks!

**Strengths And Weaknesses:**

**Strengths**

This paper considers a good range of (non-adaptive) attacks and achieves improved detection results. The difference in XAI features before and after the attacks are also clearly presented with a nice figure. Good job!

**Weakness**

The main weakness, which is also a critical one, is the lack of analysis against adaptive attacks. As the field of adversarial machine learning have advanced over years, it is imperative to have some basic checks against adaptive attacks, i.e., the attacker knows the defense mechanism in place and considers the defense in its objective. (For example, See https://proceedings.nips.cc/paper/2020/file/11f38f8ecd71867b42433548d1078e38-Paper.pdf).

As an example, I wonder
1) Are the changes in XAI features differentiable with respect to changes in the video, and
2) If yes, can you include that change in the attackers' objective function? E.g., in PGD, the attacker can introduce a negative norm of change in XAI feature in the objective as a regularization. The regularizer can also have a hyperparameter of weight: larger weight on the regularizer -> the attacker wants to keep XAI closer to the original.

---

### Review · Reviewer_Xifk · 2024-04-16

**Summary Of Contributions:**

This paper proposes to leverage XAI techniques for detecting adversarial attacks against Deepfake detectors. Specifically, they train an adversarial attack detector with PGD adversarial examples, and the model input is the concatenation of the face crop and the associated XAI map. The authors evaluate their approach on multiple white-box and black-box attacks, and examine several XAI techniques. The results show that leveraging XAI maps significantly improves the adversarial attack detection accuracy.

**Audience:**

Yes

**Broader Impact Concerns:**

No concerns.

**Claims And Evidence:**

Yes

**Requested Changes:**

1. Please add white-box attack results where the attacker has access to both the deepfake detector and the attack detector.

2. Please explain whether the attack detector can transfer across different backbone models.

3. Please add discussion on the transferability across different $\epsilon$, and explain what is the norm of the constraint. Also, please add discussion on the transferability with different attack types used for training.

4. Please add more discussion and explanation on the comparison across different XAI techniques. Also, it would be good to add results on more diverse white-box and black-box attacks, so we can see whether G-backprop is still overall better than other XAI techniques.

5. Please revise Section 5 so that XAI techniques such as G-Backprop are not referred to as attacks.

**Strengths And Weaknesses:**

Strengths:

1. The perspective of using XAI maps for adversarial attack detection is interesting.

2. The experimental results are decent, and the evaluation on XAI techniques is comprehensive.

Weaknesses

1. I appreciate the discussion of limitations in the paper, but I think more experiments are needed to justify the effectiveness of the approach. First, it is important to include white-box attack results where the attacker has access to both the deepfake detector and the attack detector. Even if such attacks are again successful, it would be helpful to see whether adding the adversarial detector makes it harder to attack.

2. Is it required to train different attack detectors for different Deepfake detectors? Have the authors evaluated the generalizability across different backbone models?

3. All adversarial examples in the evaluation share the same perturbation constraint. Have the authors evaluated the transferability across different $\epsilon$? What is the norm of the constraint, is it $L_\infty$ or $L_2$? Also, if the model is trained on other attack types, does it affect the transferability?

4. Note that while G-backprop performs better than other XAI techniques on PGD attacks, it performs much worse on Square. It would be good to add more discussion and explanation on this comparison. Also, it would be good to add results on more diverse white-box and black-box attacks, so we can see whether G-backprop is still overall better than other XAI techniques.

5. The terminology in Section 5 is confusing. G-Backprop and other XAI techniques are not attacks, they are defenses against adversarial attacks. However, Section 5 frequently refers to them as attacks.

---

### Review · Reviewer_uJpt · 2024-06-13

**Summary Of Contributions:**

The paper introduces a novel methodology for detecting adversarial attacks on deepfake detectors using eXplainable Artificial Intelligence (XAI). The primary contribution is leveraging XAI to generate interpretability maps, enhancing transparency and understanding of AI model decisions. These maps are combined with a pre-trained feature extractor to develop a robust adversarial detection model. The approach maintains the performance of existing deepfake detectors while improving their resistance to adversarial attacks. The study demonstrates promising results, suggesting potential pathways for future deepfake detection mechanisms.

**Audience:**

Yes

**Broader Impact Concerns:**

- Ethical Use of XAI: While the paper highlights the benefits of using XAI to improve model interpretability and robustness, it should also discuss potential ethical concerns related to the misuse of XAI in adversarial contexts. For example, XAI techniques could be used to reverse-engineer and exploit vulnerabilities in AI systems.

- Societal Implications: Deploying enhanced deepfake detectors can have significant societal implications, such as preventing misinformation and protecting digital content's integrity. However, there is also a potential for surveillance and privacy invasion misuse. The paper should address these dual-use concerns and propose guidelines for responsible deployment.

**Claims And Evidence:**

Yes

**Requested Changes:**

Detailed Computational Analysis:
- Provide a comprehensive analysis of the computational overhead introduced by integrating XAI techniques, particularly in real-time applications. This will help assess the practicality of deploying the proposed method in real-world settings.

**Strengths And Weaknesses:**

Strengths:
- Innovative Approach: Integrating XAI techniques to detect adversarial attacks on deepfake detectors is a novel approach that enhances transparency and interpretability. This is crucial for building trust in AI systems, especially in sensitive applications.
- Empirical Validation: The paper provides robust empirical evidence demonstrating the proposed method's effectiveness in defending against familiar and previously unseen adversarial attacks. This showcases the method's resilience and versatility.
- Minimal Performance Impact: The proposed approach does not negatively impact the performance of the deepfake detectors. This is a significant advantage as it ensures that the original detection capabilities are retained while adding an extra layer of security.
- Comprehensive Evaluation: The study includes a thorough evaluation using various attack methods and datasets, which strengthens the validity of the results.

Weaknesses:
- Focus on White-Box Attacks: The study predominantly focuses on white-box attacks, where the attacker has full access to the model. Real-world scenarios often involve black-box attacks, which are not as extensively covered.
- Assumptions About Attackers: The study assumes that attackers do not have simultaneous access to deepfake and adversarial detection mechanisms. This assumption may not always hold true in real-world settings, potentially limiting the applicability of the results.
- Dependence on XAI Maps: The proposed method's effectiveness relies heavily on the quality and reliability of the XAI maps, which may vary across different datasets and types of attacks.

---

### Decision · Action_Editor_awLW · 2024-08-16

**Recommendation:** Accept as is

**Comment:**

The reviewers pointed out that:
- The proposed methodology is innovative and shows significant promise in addressing a critical issue in deepfake detection.
- The new results and discussion after the initial review period are appreciated and they address the technical questions raised.
- This work provides an interesting perspective for Deepfake attack detection.

The overall sentiment of reviewers is positive.

**Audience:**

The paper presents an interesting perspective on XAI and deepfakes. As a result, it is interesting for the TMLR audience.

**Claims And Evidence:**

The paper states the following claims:
- an innovative approach to identifying adversarial attacks on deepfake detection systems;
- integrating XAI techniques to enhance transparency and interpretability in detecting adversarial attacks on deepfake detectors.

The authors provided enough empirical evidence to support their claims. After the rebuttal period, they added additional results as requested by the reviewers.